# Tensor Program Optimization with Probabilistic Programs

**Junru Shao**
OctoML
jshao@octoml.ai

**Xiyou Zhou**
OctoML
xiyou@octoml.ai

**Siyuan Feng**
Shanghai Jiao Tong University
hzfengsy@sjtu.edu.cn

**Bohan Hou**
Carnegie Mellon University
bohanhou@cs.cmu.edu

**Ruihang Lai**
Carnegie Mellon University
ruihangl@cs.cmu.edu

**Hongyi Jin**
Carnegie Mellon University
hongyij@cs.cmu.edu

**Wuwei Lin**
OctoML
wlin@octoml.ai

**Masahiro Masuda**
OctoML
mmasuda@octoml.ai

**Cody Hao Yu**
Amazon Web Services
hyuz@amazon.com

**Tianqi Chen**
Carnegie Mellon University, OctoML
tqchen@cmu.edu
tqchen@octoml.ai

## Abstract

Automatic optimization for tensor programs becomes increasingly important as we deploy deep learning in various environments, and efficient optimization relies on a rich search space and effective search. Most existing efforts adopt a search space which lacks the ability to efficiently enable domain experts to grow the search space. This paper introduces MetaSchedule, a domain-specific probabilistic programming language abstraction to construct a rich search space of tensor programs. Our abstraction allows domain experts to analyze the program, and easily propose stochastic choices in a modular way to compose program transformation accordingly. We also build an end-to-end learning-driven framework to find an optimized program for a given search space. Experimental results show that MetaSchedule can cover the search space used in the state-of-the-art tensor program optimization frameworks in a modular way. Additionally, it empowers domain experts to conveniently grow the search space and modularly enhance the system, which brings 48% speedup on end-to-end deep learning workloads.

## 1 Introduction

Deep learning has become pervasive in daily life. From video understanding [28], natural language understanding [15], and recommendation system [29] to autonomous driving [21], different deep learning models are deployed on different hardware platforms and devices. Deep learning frameworks usually rely on manually optimized libraries [13, 22] to accelerate deployment. Engineers need to choose from many tensor programs that are logically equivalent but differ significantly in performance due to memory access, threading, and the use of specialized hardware primitives. The engineering effort required for tensor program optimization has become a significant bottleneck for machine learning deployment with the growing number of models and hardware backends.

36th Conference on Neural Information Processing Systems (NeurIPS 2022).

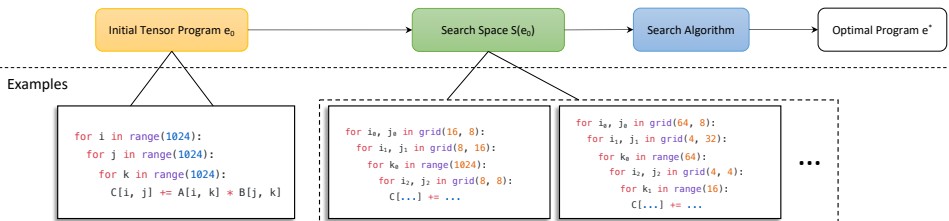

Figure 1: Automatic tensor program optimization contains two key elements: the search space $S(e_0)$ and the search algorithm that finds the optimal tensor program $e^\star$. The search space usually incorporates choices over loop transformation, vectorization, threading patterns, and hardware acceleration.

Automatic program optimization [10, 43, 1] is a recent sequence of efforts that aims to use machine learning to solve this problem. There are two vital elements of automatic tensor program optimizations. First, a search space is defined to provide a possible set of equivalent tensor programs. Then existing systems use learning-based search algorithms to find an optimized tensor program in the search space with feedback from the deployment environment. Most of the current approaches [10, 43, 1, 25, 4] use pre-defined search spaces that effectively encode the domain knowledge of the authors *once* and focus on developing efficient search algorithms.

While efficient search is essential, the search space itself fundamentally limits the best possible performance search algorithms can get. To construct a good search space, domain experts have to make numerous choices over loop transformation, vectorization, threading patterns, and hardware acceleration. Additionally, the best search space itself evolves as new tensor program optimization techniques [24] and hardware primitives [30] grow. As a result, there is a strong need to enable easy customization and construction of the search space at scale by taking inputs from system engineers and domain experts. Unfortunately, any change to search space construction currently requires surgical modifications to the automatic program optimization frameworks.

This research asks the following question: *can we decouple the search space construction from search and provide an adequate abstraction for domain experts and the learning system to collaborate on search space construction?* We give an affirmative answer to the question with two key observations. First, we can parameterize an optimization search space by the initial program followed by a sequence of transformations on the program. Next, using this parameterization, domain experts can then provide probabilistic choices that represent possible transformations after examining the program state. These two observations lead to a simple yet powerful abstraction for search space construction through a domain-specific probabilistic language. Finally, our framework composes multiple possible probabilistic transformations to form a rich search space. We make the following contributions:

- We introduce a simple yet powerful probabilistic language abstraction for tensor program search space construction.

- We build a learning-driven framework to find optimized tensor programs specified by the search space constructed using our abstraction.

- We build an end-to-end system that can take prior knowledge from domain experts to construct optimization search space to optimize deep learning deployment on multiple platforms.

Our end-to-end system can easily expand search space that matches previous approaches without surgical changes and achieve comparable performance on multiple hardware backends. Experimental results show that our abstraction is expressive enough to cover the optimization space of a diverse set of tensor programs, delivering a competitive performance of popular deep learning models, and convenient to incorporate hardware-specific knowledge into the search space to outperform the state-of-the-art frameworks.

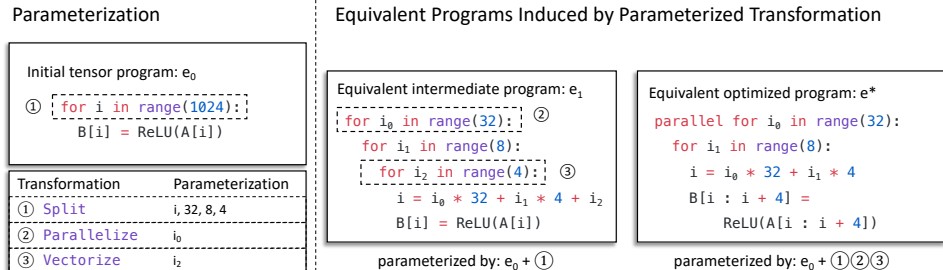

Figure 2: Parameterizing programs with the initial program and sequence of transformations. Tensor program $e_1$ is parameterized by the initial program $e_0$, plus Step ①, which is further parameterized by loop $i$ and resulting loop extents 32, 8, 4, respectively.

## 2 Background and Problem Overview

Figure 1 shows a typical workflow for tensor program optimization. For a given program $e_0$, a typical tensor program optimization framework will generate candidates from a pre-defined search space $S(e_0)$ containing semantically-equivalent programs. Then the framework finds optimized tensor program $e^* \in S(e_0)$ with the minimum latency on the target hardware.

A typical search space $S(e_0)$ contains choices over threading, loop ordering, memory access, and hardware primitives. Defining the search space $S(e_0)$ for a wide range of tensor programs brings several challenges. First, $S(e_0)$ is highly dependent on $e_0$. For example, $S(e_0)$ of a compute-intensive program (e.g., Dense) needs to consider many more possible configurations than a communication-intensive program such as ReLU. The space also differs significantly in different hardware domains. For example, $S(e_0)$ on CPU involves multi-core parallelism and vectorization, while $S(e_0)$ on GPU involves thread binding and tensorization. Finally, as the hardware and model settings change, we need to bring in fresh domain experts to update the $S(e_0)$ to leverage the latest improvements.

This paper aims to provide a programmable abstraction to construct $S(\cdot)$ in a composable and modular way. Our key goals are listed as follows: **Expressiveness.** We need to be able to build a rich search space that covers the optimization programs that domain experts will write. **Modularity.** Tensor program optimization likely will involve inputs from multiple domain experts over different periods. Therefore, we need to be able to combine prior knowledge in a composable and modular way. **Designed for learning.** We need to build a generic learning-based framework to enable diverse variations of the cost model and search for search space specified in our abstraction. We will address the above goals in the following two sections.

## 3 Composable Search Space Parameterization

This section presents MetaSchedule, a probabilistic approach to search space parameterization.

### 3.1 Stochastic Search Space Construction

MetaSchedule constructs a search space $S(\cdot)$ with stochastic program transformations as the primitive building blocks. Traditional program optimization can usually be represented by a sequence of *transformations* $\tau$, where at step $i$, the program $e_{i-1}$ is transformed into a semantically-equivalent program $e_i$, which finally leads to the optimized program $e_n$. MetaSchedule generalizes this idea by allowing further parameterization of each transformation step in $\tau$.

Taking Figure 2 as an example: $e_0$ is the initial program for the program $B = \text{ReLU}(A)$[1]. In MetaSchedule, transformation $t_1 = \text{Split}$ is parameterized by a loop $i$ and a sequence of integers indicating the loop extents after splitting; Similarly, transforms $t_2 = \text{Parallelize}$ and $t_3 = \text{Vectorize}$ are parameterized by loops respectively. As a result, an optimized program $e_n$ is obtained by applying a sequence of parameterized transformations $\tau$ to the initial program $e_0$. Accordingly, the search space $S(e_0)$ is composed of $e_0$ and all possible sequences of parameterized transformations.

---

[1]In practice, we ingest models from PyTorch/TensorFlow/JAX. See Appendix A.6 for details.

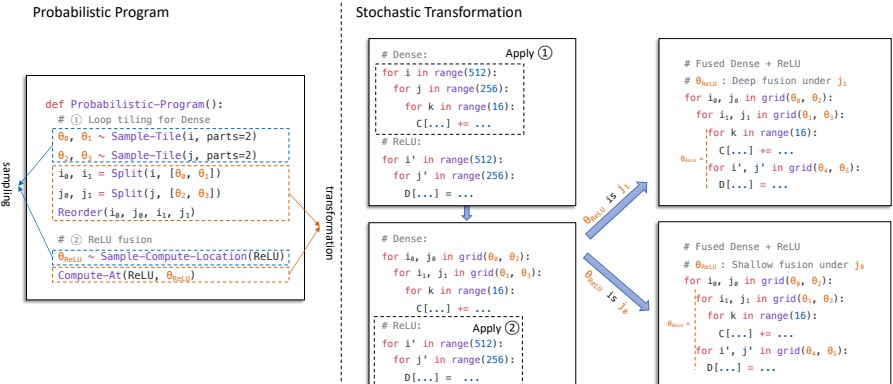

Figure 3: The MetaSchedule probabilistic language. The language contains two key elements: (1) sampling of random variables; (2) program transformation based on random variables. An example execution instance: Step ①: Draw tile sizes of and then organize the loops into a two-level tiling structure. Step ②: Decide where to fuse the `ReLU` operator.

On the other hand, it could be less practical for practitioners to determine the best combination of the parameter values in transformations. For instance, in Figure 2, it is usually efficient to use 512-bit vectorization over the inner loop when AVX-512 [2] vector instructions are available on Intel CPUs, or other vectorization lengths may lead to better performance, otherwise. Therefore, deep knowledge of the target hardware is mandatory to enumerate plausible parameter combinations to control the search space size while covering the optimal program.

To let practitioners efficiently define parameterized transformations without worrying about candidate values, MetaSchedule introduces *random variables* drawn from *analysis, sampling*. Parameterized by random variables, a transformation naturally becomes stochastic, and the underlying probabilistic space reflects the space of possible transformations.

As illustrated in Figure 3, when creating `Split` transformations to tile loop $i$ and $j$ in the `Dense` operator, the tile sizes are drawn by random variables $\theta_{0-3}$ defined from `Sample-Tile`. In this way, the `Split` transformation becomes stochastic. Similarly, we use `Sample-Compute-Location` to enumerate valid loops in `Dense` after splitting for `ReLU` to fuse its computation. In summary, 7 lines of MetaSchedule program covers a family of possible optimized tensor programs with stochastic transformations in its search space $S(e_0)$, where $e_0$ is `Dense-ReLU`.

Notably, unlike orthogonal grid space in hyperparameter tuning, MetaSchedule captures long-term structural and arithmetic dependency between random variables and the tensor program $e_i$ being transformed. As demonstrated on Step ② in Figure 3, sampling distribution from `Sample-Compute-Location` depends on the latest tensor program $e_5$, whose structure depends on all previous random variables.

## 3.2 Modular Search Space Composition

Although the search space constructed by stochastic transformations proposed in the previous subsection is efficient and is capable of covering the optimal tensor program, it is hard for other developers to learn how the search space is constructed by reading a long sequence of transformations. It makes transformations designed for a workload hard to be reused by other workloads. Meanwhile, we observe that it is straightforward to group a sub-sequence of transformations for a particular fine-grained optimization. For example, some transformations implement multi-level tiling for better memory locality in compute-intensive operators like `Conv2d` and `Dense`; some other transformations are used to fold/inline elementwise operations such as activation functions into their predecessors or successors for better memory bandwidth efficiency.

To improve the usability and make MetaSchedule more practical, we introduce *transformation module*. Just like the convolutional module with `Conv2D`, `BiasAdd` and `ReLU` in ResNet, a transformation module in MetaSchedule is defined as either atomic stochastic transformation, or composition of

---

[2] A single X86 instruction that performs the computation to a 512 bits vector in one CPU cycle.

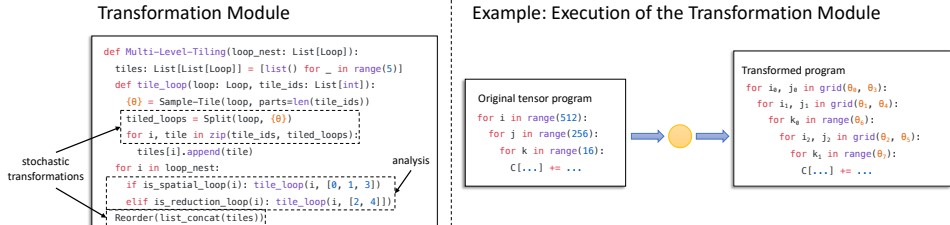

Figure 4: Transformation modules. A transformation module consists of tensor program analysis, sampling, and stochastic transformations. The figure uses `Multi-Level-Tiling` as an example. where analysis is done interactively to identify spatial (data parallel) and reduction loops, and then apply `Split` with the tiling factors drawn from `Sample-Tile`. A final `Reorder` organizes the loops into proper tiles.

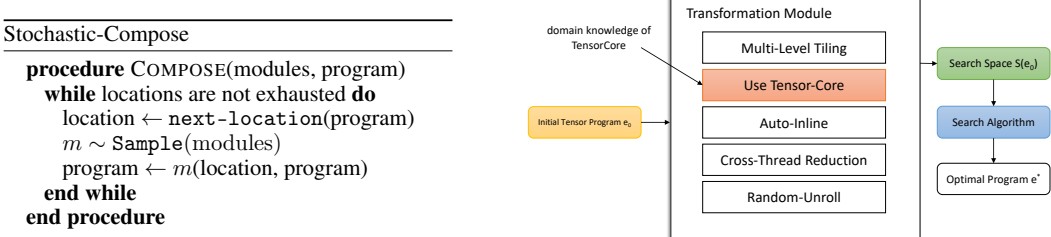

Figure 5: Left: An example algorithm to compose transformation modules. A sequence of transformation modules is composed together into a single transformation module. Right: Hierarchical composition of transformation modules gives generic search space. In this example, a hardware-specific module `Use-Tensor-Core` is composed together with other generic modules into a module that generates search space for any tensor program.

program analysis, sampling as well as smaller transformations. Each transformation module can have a meaningful name so that it can be easily adopted by many workloads to hierarchically construct a search space.

Figure 4 shows hierarchical composition of transformation modules. Specifically, `Multi-Level-Tiling` interleaves program analysis on loops and the stochastic tiling of the loop structure and organizes the original tensor program into a 5-level tiling structure. Notably, the transformation module is generic to tensor programs and thus could be applied to a variety of operators, including `conv1d`, `conv3d`, `matmul`, etc.

Figure 5 depicts an example of composing a search space with transformation modules. In this simple example, we select a set of transformation modules, which are implemented in advance by practitioners with prior domain knowledge, and apply them to every available location in the tensor program to form a search space. Consequently, the formed search space covers common optimizations on diverse hardware.

## 3.3 Relation to Existing Tensor Program Optimization Methods

In this subsection, we discuss prior approaches for automatic tensor program optimization and illustrate that many of them can be covered by the MetaSchedule framework.

**Domain specific languages for program transformations** used by prior frameworks [32, 5, 9, 37] allow developers to easily optimize a program manually. When there is no random variable sampling in the program, MetaSchedule reduces to a DSL for deterministic program transformations and achieves the same functionality.

**Template-guided auto-tuning** [10, 2, 25, 27] fully relies on developers to define a search space. In MetaSchedule, it means all random variables in a search space are defined ahead of the transformations, so there is no interaction between program analysis and follow-up random sampling choices conditioned on the program state.

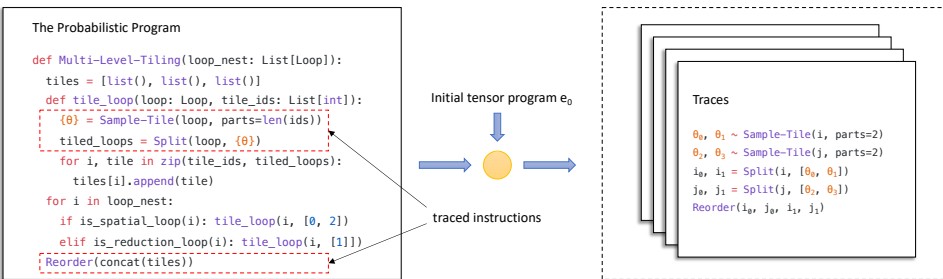

Figure 6: Execution tracing in MetaSchedule. The probabilistic language on the left defines the entire search space $S(e_0)$. Tracing the program execution across different runs leads to a set of linearized probabilistic programs on the right. Only sampling and transformation instructions are traced, while all other constructs and control flow in the host language is ignored.

**Auto-scheduling** [43, 45, 1, 19] requires developers to implement workload agnostic transformation rules. MetaSchedule achieves the same programmability and functionality through specific probabilistic transformation modules that correspond to the search space generation rules.

Notably, all approaches mentioned above have important use-cases in tensor program optimizations, depending on how much domain knowledge we want to incorporate for a particular scenario. By decoupling the search space construction from the search, we effectively build a single framework for all the use cases and enable further customization without surgical changes to the system.

## 4 Learning-driven Search

The last section provides a modular abstraction for search space. We still need to do an effective search to find an optimized program within the search space. This section provides a generic learning-driven framework to find an optimized program.

**Objective formalization.** For a given probabilistic program $e_0$, let us use $\tau$ to denote the transformations performed on $e_0$. $\tau$ can be sampled from a prior distribution specified by the probabilistic program. We define $g(e_0, \tau)$ to be the tensor program after applying transformation $\tau$ to $e_0$. Let $f(e)$ be the latency of the particular program $e$ on the hardware environment. We define a posterior probability of an optimized program as:

$$P(\tau \mid e_0) \propto e^{-f(g(e_0, \tau))} \cdot P(\tau). \tag{1}$$

Intuitively, we want to assign a higher probability to the programs that perform well. Our final goal is to find $\tau^\star = \mathrm{argmax}_\tau P(\tau \mid e_0)$ that maximizes the posterior through maximum a posteriori estimation (MAP) estimation.

**Execution tracing.** To enable domain experts to express their knowledge via transformations modules productively, we embed MetaSchedule in Python. We introduced execution tracing to reduce the cost of repetitive re-execution of the Python program. Figure 6 demonstrates an example tracing process. During program execution, the system records all samplings and transformations while ignoring control flow and other constructs of the host language. The resulting trace is a sequence of MetaSchedule primitives with only sampling and transformation instructions, which could be re-executed as a normal MetaSchedule program. We can then continue to explore different sampling choices for a given collection of initial traces. Conceptually, this is equivalent to dividing up our support set and then sampling the program condition on the execution sequence of the program.

**End-to-end search.** Figure 7 shows the overall workflow of our learning-driven framework. The search algorithm first samples the MetaSchedule program to obtain a collection of traces. Then it continues to explore the space condition on the traces. Notably, there is a significantly higher cost measuring $f(e)$ directly on the hardware, so we also incorporated a proxy cost model $\hat{f}(e)$, which is updated throughout the process, similar to previous efforts on tensor program optimization [10, 43]. At each iteration, we adopt an evolutionary search algorithm that proposes a new variant of the trace by mutating the random variables, then accept or reject the proposal based on the cost model. While evolutionary search could be viewed as parallel chain MCMC, we also made our system modular enough to incorporate other ways to select the probabilistic choices, such as those through Bayesian optimization and reinforcement learning.

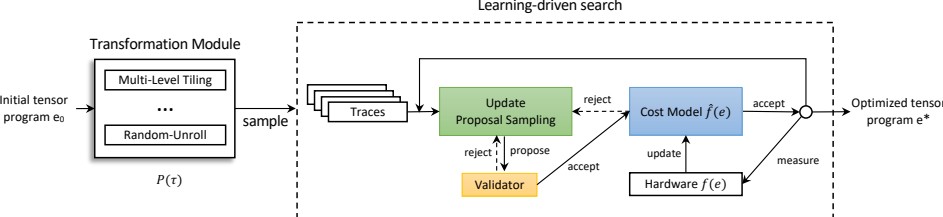

Figure 7: Learning-driven search. Based on the traces of MetaSchedule execution, candidate tensor programs are proposed by mutating sampling decisions in traces, among which the invalid ones are rejected by the validator. In every iteration, proposed candidates are accepted or rejected via annealed Metropolis-Hastings with a prediction from a learned cost model $\hat{f}$, while the cost model $\hat{f}$ is incrementally updated according to $f$, the measured latency of tensor programs on real hardware.

**Cost model.** Our approach allows extensive cost models, enabling us to supply those pre-trained from existing datasets [44]. We pick a tree-boosting-based cost model in $\hat{f}(\cdot)$ by default and leverage a common set of features that are used in previous works [43].

**Trace validation.** Importantly, invalid traces may show up as we propose updates. Such a scenario can happen when some of the random variable choices go beyond the physical hardware limit or a variable that induces changes to the execution sequence. Instead of enforcing a conservative proposal, we introduce a validator that validates the correctness of the trace. The trace validation allows us to move around the space more freely while still ensuring the correctness of the sample outcome to be on the right support set.

# 5  Related Work

Tensor Program Transformations are proposed by many prior works, such as Halide [32], TVM [9], Tiramisu [5] and TACO [23, 37]. Note that all the previous transformation languages are deterministic and cannot be directly used for search space construction, meaning that they have to introduce a separate programming model to express a search space. This paper makes a simple but powerful generalization to domain-specific probabilistic language. The resulting abstraction enables a unified approach to deterministic transformation and search space construction.

Black-box optimization has been adopted in high-performance computing libraries [16, 3]. Recent advances in automatic tensor program optimization brought a rich set of techniques to accelerate search through better cost modeling [10, 4, 34] and learning-based search [2, 25, 1, 19, 44], which could be incorporated into MetaSchedule search. Different variations of pre-defined search spaces have also been proposed that couple with the automatic tensor program optimization frameworks [10, 43, 1]. Polyhedral model [40, 6, 39] is one useful way to construct a rich pre-defined search space. This paper focuses on modular search space construction and provides orthogonal contributions to these prior works.

Probabilistic programming language is a powerful abstraction for incorporating domain knowledge and probabilistic inference. There are many general-purpose probabilistic languages, such as Church [17], Stan [8], Pyro [7], NumPyro [31], PyMC3 [35] and Edward [38]. This paper proposes a domain-specific probabilistic language for tensor program optimization with specializations such as tensor program analysis that would otherwise be opaque to the previous systems. Our learning-driven search can be viewed as an application of previous works [42, 33, 41, 46] that use tracing to divide programs into subprograms with fixed support. We focus on the MAP inference problem where the posterior depends on an unknown cost function. We solve the problem through a learned cost-model-driven evolutionary search over traces and with validation.

Automatic neural program synthesis [12, 18, 11] has seen large progress recently. Alphacode [26] builds a system that can output creative and sound solutions to problems that require deeper-level human understanding. These approaches generate abstract syntax trees (ASTs) that can be incorrect and use input-output pairs to filter out those erroring programs. Our compiler approach requires us to ensure the correctness of all transformations. However, some ideas like validation after creation might be reusable.

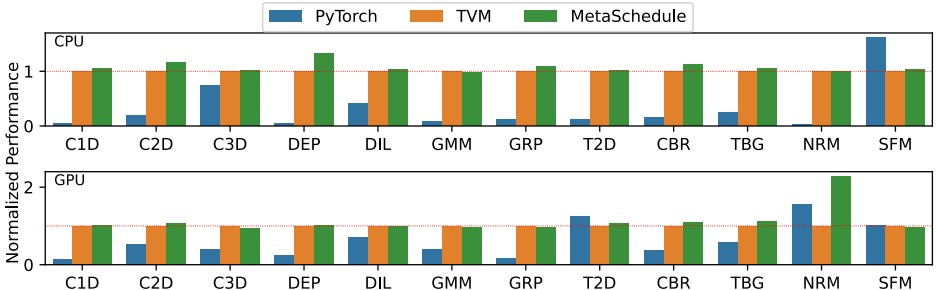

Figure 8: Operator- and subgraph-level performance. MetaSchedule always achieves similar or better performance compared with TVM (Ansor), and exceeds PyTorch significantly in most workloads whose operators are carefully hand-optimized.

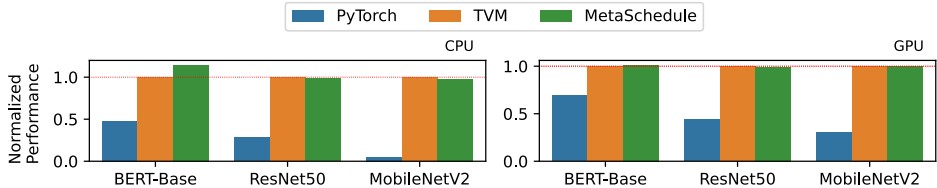

Figure 9: Optimizing end-to-end deep learning models. MetaSchedule reaches close or better performance compared to TVM (Ansor) on all the models.

## 6 Experiments

### 6.1 Expressiveness to Cover Common Optimization Techniques

This section aims to answer the following question: *Is MetaSchedule expressive enough to capture the search space of the state-of-the-art optimization techniques?* To answer this question, we evaluate our work on a diverse set of operators and subgraphs extracted from popular deep learning models, including variants of convolution, dense, and normalization.

As baselines, PyTorch (v1.11.0) results are provided to compare performance with vendor libraries; TVM (commit: 8d4f4dd73f), which incorporates AutoTVM [10] and Ansor [43], is used as the state-of-the-art tensor program optimization system, and we pick the best among the two in each respective setups. Full operators and hardware configurations are documented in Appendix A.2.

Figure 8 shows that, in all cases on CPU and GPU, MetaSchedule delivers performance comparable with or even better than TVM, from which we could infer that MetaSchedule could express optimization techniques comparable to TVM on diverse workloads. Additionally, in most of the cases, MetaSchedule outperforms PyTorch by a significant margin except for SFM, which is highly optimized manually in PyTorch.

### 6.2 Optimizing End-to-End Deep Learning Models

Operator performance does not always translate directly to full model optimization. Therefore, this section is dedicated to answering the following question: *Can MetaSchedule deliver competitive performance with state-of-the-art works for end-to-end models?*

Therefore, a series of experiments are conducted to compare MetaSchedule and TVM, including BERT-Base [14], ResNet-50 [20], and MobileNet-v2 [36] on both CPU and GPU. As shown in Figure 9, MetaSchedule performance is on parity with TVM, while surpassing PyTorch in all cases, which indicates that MetaSchedule framework delivers end-to-end performance. Additionally, tuning time is provided in Appendix A.5.

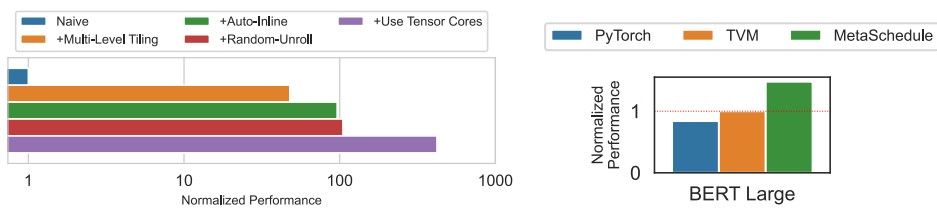

(a) Performance with different search spaces.
(b) BERT-Large Performance.

Figure 10: Left: Search space composition conducted on a representative subgraph of BERT called `fused-dense`. Gradually composing more transformation modules, the space covers more optimized programs. Right: Introduction of hardware-specific `Use-Tensor-Core` module, composed with existing search space, brought 48% speedup over TVM (AutoTVM).

### 6.3 Search Space Composition and Hardware-Specific Modules

Besides performance parity with existing work, in this section, we demonstrate the extra value of modular search space composition by answering the following question: *How convenient is it to compose transformation modules, and how does it translate to performance?*

We design an ablation study for transformation modules composition. As indicated in Figure 10a, by progressively enriching the search space, the performance of optimized tensor programs consistently increases. Composed with a hardware-specific module `Use-Tensor-Core`, MetaSchedule delivers significantly better performance compared with generic search space. The performance gain, brought by search space composition with customized rules, does translate to end-to-end model performance, as shown in Figure 10b. Specifically, on BERT-large workloads, MetaSchedule with `Use-Tensor-Core` delivers 48% speedup over TVM.

Notably, it took a graduate student only 2 days to craft the 82-line `Use-Tensor-Core` module in Python (see supplementary materials), which provides strong evidence of the convenience of customization and composition. More details are in Appendix A.4.

## 7 Conclusion

This paper presents MetaSchedule, a programming model to describe search space construction in tensor program optimization. Our method abstracts search space as a probabilistic language and enables flexible incorporation of domain knowledge by allowing practitioners to implement customized probabilistic programs. A learning-driven search algorithm is developed on top of the probabilistic language abstraction, which delivers competitive performance with state-of-the-art frameworks. In the future, we will explore and modularize declarative API for various hardware environments. Therefore, we will open-source our framework and hope it could enable broader collaboration between the machine learning deployment engineers and intelligent machine learning algorithms for tensor programs.

## Acknowledgement

This work is supported in part by a gift from Oppo. We would like to thank Josh Fromm, Denise Kutnick and Sunghyun Park from OctoML for helpful discussion and support of the work.

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
