# A Appendix

## A.1 Environment Setup for Experiments

All CPU experiments are conducted on AWS C5.9xlarge instances with Intel Xeon Platinum 8124M CPUs. All GPU experiments are done with NVIDIA GeForce RTX 3070 graphics cards.

## A.2 Workload Configurations in the Evaluation

- C1D (1D Convolution): batch=1, length=256, input channel=64, output channel=128, kernel size=3, stride=2, padding=1
- C2D (2D Convolution): batch=1, height=224, width=224 input channel=3, output channel=64, kernel size=7, stride=2, padding=3
- C3D (3D Convolution): batch=1, depth=16, height=224, width=224 input channel=3, output channel=64, kernel size=7, stride=2, padding=3
- DEP (Depthwise Convolution): batch=1, height=112, width=112 channel=32, kernel size=3, stride=1, padding=1
- DIL (Dilated Convolution): batch=1, height=224, width=224 input channel=3, output channel=64, kernel size=7, stride=2, padding=3, dilation=2
- GMM (Matrix Multiply): batch=1, N=M=K=128
- GRP (Group Convolution): batch=1, height=56, width=56 input channel=64, output channel=128, kernel size=3, stride=2, padding=1, groups=4
- T2D (Transposed 2D Convolution): batch=1, height=4, width=4 input channel=512, output channel=256, kernel size=4, stride=2, padding=1
- CBR (2D Convolution + Batch Norm + RuLU): batch=1, height=224, width=224 input channel=3, output channel=64, kernel size=7, stride=2, padding=3
- TBG (Transpose + Matrix Multiply): batch=1, seq=128, head=12, dim=64
- NRM (Norm): batch=1, m=256, n=256
- SFM (Softmax): batch=1, m=256, n=256

## A.3 Use-Tensor-Core Search Space Definition Code

```
b0 = sch.get_block(name="T_dense", func_name="main")
b1 = sch.get_block(name="T_add", func_name="main")
b2 = sch.get_block(name="T_multiply", func_name="main")
b3 = sch.get_block(name="T_cast", func_name="main")
b4 = sch.get_block(name="T_erf", func_name="main")
b5 = sch.get_block(name="T_cast_1", func_name="main")
b6 = sch.get_block(name="T_multiply_1", func_name="main")
b7 = sch.get_block(name="T_add_1", func_name="main")
b8 = sch.get_block(name="T_multiply_2", func_name="main")
b9 = sch.get_block(name="root", func_name="main")
sch.annotate(block_or_loop=b0, ann_key="tiling_structure", ann_val="SSSRRSRS")
b10 = sch.reindex(block=b0, buffer_index=0, is_write_index=True)
b11 = sch.reindex(block=b0, buffer_index=0, is_write_index=False)
b12 = sch.reindex(block=b0, buffer_index=1, is_write_index=False)
sch.transform_block_layout(block=b10, index_map=lambda i_l, j_l, k_l: (i_l, j_l, k_l, ))
sch.transform_block_layout(block=b11, index_map=lambda i_l, j_l, k_l: (i_l, j_l, k_l, ))
sch.transform_block_layout(block=b12, index_map=lambda i_l, j_l, k_l: (i_l, j_l, k_l, ))
sch.transform_block_layout(block=b0, index_map=lambda i_l, j_l, k_l: (i_l, j_l, k_l, ))
sch.transform_layout(block=b0, buffer_index=0, buffer_index_type=write, index_map=lambda i_l, j_l: (i_l, j_l, ))
sch.transform_layout(block=b0, buffer_index=1, buffer_index_type=read, index_map=lambda j_l, k_l: (j_l, k_l, ))
sch.transform_layout(block=b0, buffer_index=0, buffer_index_type=read, index_map=lambda i_l, k_l: (i_l, k_l, ))
l13, l14, l15 = sch.get_loops(block=b0)
l16, l17 = sch.split(loop=l15, factors=[64, 16])
l18, l19 = sch.split(loop=l14, factors=[256, 16])
l20, l21 = sch.split(loop=l13, factors=[64, 16])
l22, l23, l24, l25, l26, l27 = sch.get_loops(block=b0)
sch.reorder(l24, l26, l21, l19, l17)
b28 = sch.blockize(loop=l21)
sch.annotate(block_or_loop=b0, ann_key="auto_tensorize", ann_val="wmma_sync")
sch.annotate(block_or_loop=b28, ann_key="auto_tensorize", ann_val="wmma_fill")
b29 = sch.get_block(name="root", func_name="main")
sch.annotate(block_or_loop=b29, ann_key="tensor_core_enabled", ann_val="1")
b30 = sch.get_block(name="root", func_name="main")
sch.annotate(block_or_loop=b30, ann_key="warp_execution", ann_val=1)
l31, l32, l33 = sch.get_loops(block=b28)
v34, v35, v36, v37, v38 = sch.sample_perfect_tile(loop=l31, n=5, max_innermost_factor=4, decision=[8, 1, 2, 2, 2])
l39, l40, l41, l42, l43 = sch.split(loop=l31, factors=[v34, v35, v36, v37, v38])
v44, v45, v46, v47, v48 = sch.sample_perfect_tile(loop=l32, n=5, max_innermost_factor=4, decision=[1, 32, 4, 2, 1])
l49, l50, l51, l52, l53 = sch.split(loop=l32, factors=[v44, v45, v46, v47, v48])
v54, v55, v56 = sch.sample_perfect_tile(loop=l33, n=3, max_innermost_factor=4, decision=[32, 2, 1])
l57, l58, l59 = sch.split(loop=l33, factors=[v54, v55, v56])
sch.reorder(l39, l49, l40, l50, l41, l51, l57, l58, l42, l52, l59, l43, l53)
l60 = sch.fuse(l39, l49)
sch.bind(loop=l60, thread_axis="blockIdx.x")
l61 = sch.fuse(l40, l50)
sch.bind(loop=l61, thread_axis="blockIdx.y")
l62 = sch.fuse(l41, l51)
sch.bind(loop=l62, thread_axis="threadIdx.y")
sch.annotate(block_or_loop=b28, ann_key="thread_extent_low_inclusive", ann_val=32)
sch.annotate(block_or_loop=b28, ann_key="thread_extent_high_inclusive", ann_val=1024)
b63 = sch.write_at(loop=l62, block=b28, write_buffer_index=0, storage_scope="wmma.accumulator")
sch.reverse_compute_inline(block=b10)
v64 = sch.sample_categorical(candidates=[4, 8, 16], probs=[0.0.33, 0.0.33, 0.0.33], decision=0)
sch.annotate(block_or_loop=b63, ann_key="vector_bytes", ann_val=v64)
b65 = sch.read_at(loop=l57, block=b28, read_buffer_index=0, storage_scope="shared.dyn")
v66 = sch.sample_categorical(candidates=[4, 8, 16], probs=[0.0.33, 0.0.33, 0.0.33], decision=2)
sch.annotate(block_or_loop=b65, ann_key="vector_bytes", ann_val=v66)
sch.annotate(block_or_loop=b65, ann_key="local_stage", ann_val=1)
sch.annotate(block_or_loop=b65, ann_key="double_buffer_scope", ann_val=0)
b67 = sch.read_at(loop=l57, block=b28, read_buffer_index=1, storage_scope="shared.dyn")
v68 = sch.sample_categorical(candidates=[4, 8, 16], probs=[0.0.33, 0.0.33, 0.0.33], decision=2)
sch.annotate(block_or_loop=b67, ann_key="vector_bytes", ann_val=v68)
sch.annotate(block_or_loop=b67, ann_key="local_stage", ann_val=1)
sch.annotate(block_or_loop=b67, ann_key="double_buffer_scope", ann_val=0)
b69 = sch.read_at(loop=l58, block=b28, read_buffer_index=0, storage_scope="wmma.matrix_a")
b70 = sch.read_at(loop=l58, block=b28, read_buffer_index=1, storage_scope="wmma.matrix_b")
sch.compute_inline(block=b11)
sch.compute_inline(block=b12)
sch.annotate(block_or_loop=l58, ann_key="software_pipeline_stage", ann_val=[0, 0, 1])
sch.annotate(block_or_loop=l58, ann_key="software_pipeline_order", ann_val=[0, 1, 2])
sch.annotate(block_or_loop=l57, ann_key="software_pipeline_stage", ann_val=[0, 0, 0, 0, 0, 1, 1])
sch.annotate(block_or_loop=l57, ann_key="software_pipeline_order", ann_val=[0, 3, 1, 4, 5, 2, 6])
sch.compute_inline(block=b7)
sch.compute_inline(block=b6)
sch.compute_inline(block=b5)
sch.compute_inline(block=b4)
sch.compute_inline(block=b3)
sch.compute_inline(block=b2)
sch.compute_inline(block=b1)
sch.reverse_compute_inline(block=b8)
v71 = sch.sample_categorical(candidates=[0, 16, 64, 512, 1024], probs=[0.2, 0.2, 0.2, 0.2, 0.2], decision=3)
sch.annotate(block_or_loop=b9, ann_key="unroll_explicit", ann_val=v71)
```

## A.4 Search Space Construction and Customization for New Hardware

Our previous experience suggests that adapting to new hardware would require months of effort and deep expertise in the compilation stack, across code generation, transformations, and search across the legacy codebase.

Take TensorCore GPUs as an example. Unfortunately, Ansor in TVM does not support TensorCore, and it is highly non-trivial to extend its support to cover TensorCore without a dramatic revamp to its IR; Similarly, AutoTVM does not deliver comparable TensorCore performance without a dramatic re-design of its schedule tree, even though it supports a limited set of tensor intrinsics (e.g. WMMA). In both cases, with the assumption of upgrading the core design, we anticipate that it will take months of engineering effort, and more to first get familiar with the codebase.

MetaSchedule enables doing that without such coupled complexity and no prerequisite experience on code generation or existing transformations, where each transformation is modeled as an independent primitive in the probabilistic language. Incorporating TensorCore, as an independent program transformation, could be engineered within 2 days by a grad student, and then composed into the existing system without having to re-design or affect any existing functionality.

## A.5 Tuning Time

MetaSchedule makes an orthogonal contribution as it is a probabilistic language for composable search space construction rather than speeding up tuning. To provide a fair comparison of tuning speed, we reproduced Ansor's search space in our MetaSchedule probabilistic language in Table 1.

|              | TVM Ansor (min) | MetaSchedule (min) |
|--------------|-----------------|--------------------|
| ResNet-50    | 287.17          | 220.66             |
| BERT-base    | 292.45          | 288.4644           |
| MobileNet-v2 | 280.53          | 251.8831           |
| GPT-2        | 270.97          | 214.1358           |
| Inception-v1 | 295.766         | 290.3632           |

Table 1: Tuning time comparison.

## A.6 End-to-End Integration Workflow with Deep Learning Framework

From frontend frameworks, for example, TensorFlow, PyTorch, or JAX, the tensor program to be optimized is generated from their computational graph. The generation process is generic to the shapes/ranks of tensors. The MetaSchedule program is generated from the tensor program obtained, based on which the search algorithm is performed.

## A.7 Available Transformations Primitives

| Transformation | Explanation |
|---|---|
| split | Split a loop into a sequence of consecutive loops |
| fuse | Fuse a sequence of consecutive loops into one |
| reorder | Reorder a sequence of loops |
| parallel | Parallelize a loop across CPU cores |
| vectorize | Vectorize a loop with SIMD |
| unroll | Unroll a loop |
| bind | Bind a loop to a GPU thread |
| cache-read | Create a block that reads a buffer region into a read cache |
| cache-write | Create a block that writes a buffer region into a write cache |
| compute-at | Move a producer block under the specific loop |
| compute-inline | Inline a block into its consumer(s) |
| rfactor | Factorize an associative reduction block by the specified loop |
| storage-align | Set alignment requirement for specific dimension of a buffer |
| set-scope | Set the storage scope of a buffer |
| add-unit-loop | Create a new unit loop on top of the specific block |
| re-index | Create a block that read/write a buffer region into a read/write cache with reindexing |
| reverse-compute-at | Move a consumer block under the specific loop |
| reverse-compute-inline | Inline a block into its only producer |
| decompose-reduction | Decompose a reduction block into two separate blocks |
| blockize | Convert the subtree rooted at a specific loop into a block |
| tensorize | Tensorize the computation enclosed by loop with the tensor intrin |
| annotate | Annotate a block/loop with a key value pair |
| unannotate | Unannotate a block/loop's |
| transform-layout | Apply a transformation to buffer layout, represented by an index map |
| transform-block-layout | Apply a transformation to block layout, represented by an index map |
| decompose-padding | Decompose a padding block into a block filling const pad values and a block writing in-bound values |
| sample-categorical | Sample an integer given the probability distribution |
| sample-perfect-tile | Sample the factors to perfect tile a specific loop |
| sample-compute-location | Sample a compute-at location of the given block |

Table 2: All available transformation primitives.