# OpenReview forum: "Tensor Program Optimization with Probabilistic Programs"
_NeurIPS.cc/2022/Conference — NeurIPS 2022 Accept_

### Official Review · Reviewer_UWHJ · 2022-07-09

**Rating:** 5
**Confidence:** 4
**Soundness:** 3 good
**Presentation:** 2 fair
**Contribution:** 3 good

**Summary:**

This paper introduces SpaceCraft, which improves the efficiency of designing tensor programs’ schedulers via establishing the search space in a composable and modular way. The key of SpaceCraft is to use the probabilistic programming method to describe the search space. Designers can improve their efficiency by combining SpaceCraft’s reusable program transformation modules with prior domain knowledge. The paper also introduces a learning-driven searching framework that can search efficiently under the guidance of a learned cost model. The experiments demonstrate that SpaceCraft achieves comparable optimization ability against TVM.

**Questions:**

Please address the weaknesses listed above. Besides, in Figure 10(b), does TVM use Tensor Core as well?

**Limitations:**

Supporting tensor computation unit is critical while challenging for existing exploration-based approached. Both FlexTensor and Ansor cannot support Tensor Core in a convenient manner. The readers would expect to see more explanation on the Use-Tensor-Core module.

**Strengths And Weaknesses:**

This paper proposes a new programming model, which abstracts the search space as a probabilistic language and allows incorporation of domain knowledge, for tensor program optimization. The proposed method is technically sound. It seems that prior wok such as AutoTVM and Ansor can be easily implemented by using SpaceCraft, and SpaceCraft works well for domain-specific accelerators such as Tensor Core. The related work, including template-guided auto-tuning and auto-scheduling, is adequately cited.

This work is evaluated in terms of both expressiveness and optimization performance. Although it achieves comparable performance against TVM, it would be quite helpful to compare the performance with state-of-the-art approaches including FlexTensor and Ansor. Moreover, as SpaceCraft can provide a better search space, I suggest to directly compare the quality of produced search spaces against previous work, for example, ratio of high-performance candidate programs or performance difference of candidate programs, etc.

The advantage of the proposed learning-driven search over existing AutoTVM is unclear, which employs the cost model as well. I would like to see more ablation study on the effectiveness of the proposed learning framework.

---

> ### Author Response · Authors · 2022-08-02
> **Thank you for your valuable reviews and suggestions**
>
> Thank you for your valuable reviews and suggestions. We make the following clarifications here:
>
> > it would be quite helpful to compare the performance with state-of-the-art approaches including FlexTensor and Ansor.
>
> We use the TVM as the baseline and turn on AutoTVM and Ansor in our evaluation (and give the better one). Specifically, we use Ansor for all fp32 workloads and AutoTVM in figure 10(Ansor does not support TensorCore). Ansor is shown to perform better than FlexTensor. So these baseline results indeed reflect the STOA. Thank you for bringing this up, and we will clarify it in the evaluation setup.
>
> > The advantage of the proposed learning-driven search over existing AutoTVM
>
> The main focus of the paper is search space construction, and is orthogonal to works that optimizes search. This being said, we do need to tailor the search to support the general probabilistic programming approach via tracing.
>
> > more explanation on the Use-Tensor-Core module
>
> Thank you for your valuable suggestion! We plan to add discussions about key elements in the use-tensor-core module.
>
> > directly compare the quality of produced search spaces against previous work, for example, ratio of high-performance candidate programs or performance difference of candidate programs
>
> Thank you for your suggestions. The search space is quite large and it is hard to enumerate all the programs (and random samples usually produce programs on lower percentiles that are less interesting). We believe the end to end results after optimization  is a good proxy to evaluate the search space.

---

### Official Review · Reviewer_AXec · 2022-07-11

**Rating:** 6
**Confidence:** 3
**Soundness:** 2 fair
**Presentation:** 2 fair
**Contribution:** 3 good

**Summary:**

Program optimization relies on rewriting a given program into an equivalent, but more efficient version of itself. These equivalent programs are searched within hand-engineered search spaces which are typically hard to customize. This paper introduces SpaceCraft, a domain-specific probabilistic-programming-language-abstraction which enables easy customization and construction of tensor program search spaces. SpaceCraft achieves this goal by providing modularity for composing program transformations. Further, the authors combine SpaceCraft with MAP estimation for automatic tensor programs optimization. SpaceCraft can cover the optimization search space used in many prior work and optimization with SpaceCraft delivers similar performance to prior work. On growing the search-space with a hardware specific module, the optimization yields a significant improvement over prior work.

**Questions:**

Can the authors can address the weaknesses raised in the previous section?

Specifically:

1. Can they address the issues with the text of the paper?
2. Can the address the drawbacks of their experimental design?
3. Can the authors explain why certain important experiments have not been conducted?

**Limitations:**

No. The authors have not discussed limitations of this approach. The authors can discuss trade-offs of using MAP estimation vs. techniques such as reinforcement learning or simulated annealing used in prior work. The authors can also highlight the cost of learning a new DSL for crafting search spaces in SpaceCraft.

**Strengths And Weaknesses:**

# Strengths:

1. The program tackles an very useful problem - construction of the program search space. Typically,  construction of the "right" search space requires many trial and errors. SpaceCraft can significantly help reduce the complexity of this process.
2. Clarity: The figures effectively convey the intended message.
3. The use of probabilistic programs for tensor program optimization is novel.
4. SpaceCraft covers the prior methods under a single framework.

# Weaknesses:

1. **Presentation**:  The presentation of the paper requires further refinement. Apart from plenty of minor grammatical mistakes (specified below), the paper also consist of many confusing sentences:
	1. The paper fails to specify what exactly are the hurdles with search space construction/customization in prior work. The paper simply states previous methods "... requires surgical modification...".  What are the surgical modifications? Why are they costly/infeasible?
	2. In Line 44-52, paper states that " These two observations lead to a simple yet powerful abstraction ..." but do not to specify what exactly this abstraction is.
	3. In Line 124-126, the paper states that SpaceCraft captures long-term structural arithmatic dependency between random variables and tensor program. The paper would benefit from specifically state why SpaceCraft achieves this.
	4. The presentation in Section 3.3 is very confusing. In this section, the reader needs to be aware of the authors intentions to understand some sentences. For example,  in Lines 163-165, what is implied by "it" is not clear.

2.  **Experiments**: There are multiple drawbacks with the experiments section in the paper:
	1. A central benefit of SpaceCraft is that it simplifies the construction and modification of the optimization search spaces. This benefit however, and not been evaluated at all. This benefit should be evaluated by comparing the human efforts required for introducing new modules in prior frameworks versus the effort required for the same in SpaceCraft.
	2. The paper states (Line 278) that it takes 2 days to introduce a new module, but what is the reader supposed to compare this number with? What is the time it would take for introducing similar modules in prior work?
	3. The paper does not compare their optimization results to newer optimization techniques such as AutoTVM [1] and CHAMELEON [2] making it unclear how well this framework fares w.r.t. the current state-of-the-art.
	4. Comparisons in Figure 10 (b) are unfair as the search space of SpaceCraft and TVM are different. Are there fundamental limitations to TVM which make expanding the search space with operations from "Use-Tensor-Core" module infeasible?
	5. All the plots in Figure 8, 9, and 10  present results with normalized performance without clarifying what is the normalized performance measures. Further, the paper would benefit if more interpret-able statistics such as wall clock time or TFLOPS are used.
	6. The paper does not contain any measurement of the time it takes to perform the proposed automatic optimization. This is important as a high optimization cost would make the usage of SpaceCraft infeasible for many situations.

## Minor Weaknesses


1. In the conclusions, the paper attempt to say they will open-source the framework. However, in the checklist, they mention the code and data is proprietary. In that case, how will they eventually open source the framework?
2.   In the checklist, the authors state that limitation of the work are described in section 7. However, there are no sentences that discuss any limitations of the work in the entire paper.


## Errors:

Line 18: has become

Line 70 : Then, the framework finds

Line 83: optimization programs

Line 84: Therefore/Hence, we need

Line 87: build a generic

Line 288: We will open-source our framework

Line 481: in the Appendix.


## References:


[1]  Learning to Optimize Tensor Programs, Tianqi Chen et al., NeurIPS 2018

[2] Chameleon: Adaptive Code Optimization for Expedited Deep Neural Network Compilation, Byung Hoon Ahn et al., ICLR 2020

---

> ### Author Response · Authors · 2022-08-02
> **Thank you for your valuable reviews**
>
> Thank you for your valuable reviews. The comments on writings are helpful and we plan to incorporate them in future versions. We make the following clarifications here:
>
> > hurdles with search space construction/customization
>
> Thanks for bringing this up! We agree that such complexity is subjective but is again an important one.  Extending search space, such as those in Ansor, would require changes to its internal search rule and search in C++, as well as the corresponding code-generation and transformation stack. We attempted to do so and found it could take months or, if not more effort. Additionally, doing such a change would require deep knowledge to learn the Ansor/TVM codebase. These experiences motivate SpaceCraft, where someone who is not necessarily familiar with the internals of the compiler can easily extend the space. We believe this is a very important step in making automated tensor program optimization accessible to a broader set of hardware providers and ML engineers.
>
> This is a great point and we plan to add discussion to our final version.
>
> > long-term structural arithmetic dependency between random variables and tensor program
>
> Line 126-128 and Figure 3 provide a demonstration of how SpaceCraft models the long-term dependency. Essentially, sampling distribution of `Sample-Compute-Location` depends on all previous random variables and the program structure they result in. We will add a note in the related discussions.
>
> >  evaluation on the overall cost of constructing search space, and cost of introducing new modules.
>
> The cost of search space construction is indeed subjective and hard to evaluate but is also a very important factor. Our previous experience suggests that making such changes would require months of effort and need deep experience of the compilation stack across code generation, transformations, and search across C++ and python. SpaceCraft enables doing that without such coupled complexity and no prerequisite experience on codegen/transformations. We will add discussions about it to emphasize the point.
>
> > compare results to newer optimization techniques such as AutoTVM  and CHAMELEON
>
> We use the TVM as the baseline and turn on AutoTVM and Ansor in our evaluation (and give the better one). Specifically, we use Ansor for all fp32 workloads and AutoTVM in Figure 10 (Ansor does not support TensorCore). So these baseline results indeed reflect the STOA. Thank you for bringing this up, and we will clarify it in the evaluation setup.
>
> > un-normalized metrics such as TFLOPS
>
> Thank you for your suggestion. We normalized the numbers mainly because the unnormalized results differ quite a lot by the nature of workloads. The normalized performance is also a common metric used by previous papers, such as those in Ansor. We agree that unnormalized results can be helpful and plan to add metrics as well in the appendix.
>
> > Comparisons in Figure 10 (b) are unfair as the search space of SpaceCraft and TVM are different.
>
> Thank you for your input. SpaceCraft can indeed generate better search space in this case than TVM. It would take months of effort and in-depth knowledge of the compiler stack to add the same search space to TVM. SpaceCraft alleviates that burden, and this is the main contribution of the paper.
>
> > In the checklist, the authors state that limitations of the work are described in section 7. However, there are no sentences that discuss any limitations of the work in the entire paper
>
> Thank you for bringing it up. As we stated in line 287 - 288 in Section 7, one limitation is that we haven’t covered hardwares other than CPU/GPUs and we plan to do them as future works.
>
> > However, in the checklist, they mention the code and data is proprietary and open source
>
> Thanks for bringing this up. We think the review probably refers to L454-455 in the checklist. Here “The code and the data are proprietary” is an example answer in NeurIPS templates, not our answer to the question. We plan to upstream our code to TVM under Apache 2.0.

---

> > ### Comment · Reviewer_AXec · 2022-08-06
> > **Thank you for the rebuttal!**
> >
> > Thank you for your replies to the issue I raised with the paper. Overall, if the final draft does contain the changes that the authors plan to add, I would be happy to raise my score to a Weak Accept. This decision is also motivated by Reviewer zFyW's insightful comments.
> >
> > My central concerns were as follows and have been addressed by the authors:
> >
> > 1. Presentation: If the authors implement the presentation changes suggested by all the reviewers, this paper would become far more accessible (and in turn, perhaps more impactful).
> >
> > 2. Comparison to SOTA: The authors clarify that comparison has been made to the state-of-the-art and will be clarified in the revisions (reviewer UWHJ had the same concern).
> >
> > 3. Evaluation: I believe some of the evaluation related questions might not be practically feasible (for example, the excessive investment required to implement the "Use-Tensor-Core" module in TVM).
> >
> > However, there are some questions the authors have not answered, and I'll be curious to get their responses:
> >
> > 1) Can the authors provide any measurement of the time it takes to perform the proposed automatic optimization?
> > 2) It takes 2 days to introduce a new module in SpaceCraft (as stated in the paper). What should the readers compare this number with? Is there at least a rough estimate of the time it would take to introduce the *same* module in TVM?

---

> > > ### Author Response · Authors · 2022-08-08
> > > **Paper revision and addressing questions**
> > >
> > > Thank you for your response! We updated the paper and make sure the relevant discussion and clarification are included.
> > >
> > > > Can the authors provide any measurement of the time it takes to perform the proposed automatic optimization?
> > >
> > > SpaceCraft makes an orthogonal contribution as it is a probabilistic language for composable search space construction rather than speeding up tuning.
> > >
> > > To provide a fair comparison of tuning speed, we reproduced Ansor’s search space in our SpaceCraft probabilistic language. Please refer to the table below for details:
> > >
> > > |              | Ansor (min) | SpaceCraft (min) |
> > > |--------------|-------------|------------------|
> > > | ResNet-50    | 287.17      | 220.66           |
> > > | BERT         | 292.45      | 288.4644         |
> > > | MobileNet-v2 | 280.53      | 251.8831         |
> > > | GPT-2        | 270.97      | 214.1358         |
> > > | Inception-v1 | 295.766     | 290.3632         |
> > >
> > > > It takes 2 days to introduce a new module in SpaceCraft (as stated in the paper). What should the readers compare this number with? Is there at least a rough estimate of the time it would take to introduce the same module in TVM?
> > >
> > > Unfortunately, Ansor in TVM does not support TensorCore, and it is highly non-trivial to extend its support to cover TensorCore without a dramatic revamp to its IR; Similarly, AutoTVM does not deliver comparable TensorCore performance without a dramatic re-design of its schedule tree, even though it supports a limited set of tensor intrinsics (e.g. WMMA). In both cases, with the assumption of upgrading the core design, we anticipate that it will take months of engineering effort, and more to first get familiar with the codebase.
> > >
> > > To compare the effort to craft tensor-core-aware high-performance GEMM operations (e.g. batch matmul and matmul), which is the backbone of BERT, we compare the SOTA open-source CUTLASS library with SpaceCraft. At the time of writing, CUTLASS has 68337 lines of code in 127 files under its `include/cutlass/gemm` folder, which is unlikely to be implemented within 2 days.
> > >
> > > Thanks to the composable design and decoupling of search space and search, SpaceCraft makes such research space construction more accessible.

---

> > > > ### Comment · Reviewer_AXec · 2022-08-09
> > > > **Thank you for the additional information!**
> > > >
> > > > Thank you very much for your comments. The list of changes are encouraging!
> > > >
> > > >
> > > > > To provide a fair comparison of tuning speed, we reproduced Ansor’s search space in our SpaceCraft probabilistic language. Please refer to the table below for details:
> > > >
> > > > Thank you for the additional information. It would be great if these results could be accessible through a supplementary or even a `README.md` in your public code release.
> > > >
> > > > > At the time of writing, CUTLASS has 68337 lines of code in 127 files under its include/cutlass/gemm folder, which is unlikely to be implemented within 2 days.
> > > >
> > > > I believe these stats can be a useful for readers unfamiliar with the domain to glean at the importance of SpaceCraft. It would be great to include this information in perhaps the paper supplementary.

---

> > > > > ### Author Response · Authors · 2022-08-09
> > > > > **Thank you for following up!**
> > > > >
> > > > > Thank you so much for your response!
> > > > >
> > > > > > It would be great if these results could be accessible through a supplementary or even a README.md in your public code release.
> > > > >
> > > > > Thank you for the valuable suggestion! Scripts will be provided to reproduce Ansor's search space using SpaceCraft on a wide range of popular models.
> > > > >
> > > > > > I believe these stats can be a useful for readers unfamiliar with the domain to glean at the importance of SpaceCraft. It would be great to include this information in perhaps the paper supplementary.
> > > > >
> > > > > That makes sense! A subsection is added in the Appendix on customization with new hardware, TensorCore, and CUTLASS.

---

### Official Review · Reviewer_zFyW · 2022-07-11

**Rating:** 6
**Confidence:** 4
**Soundness:** 3 good
**Presentation:** 2 fair
**Contribution:** 4 excellent

**Summary:**

SpaceCraft is developed as an improvement over previous TVM IR such that it can unlock two key features: (1) not having to define the template like initial AutoTVM and its variants, and (2) achieving same programmability as Auto Scheduling such as Ansor. Overall, the paper devise a new abstraction with probabilistic language abstractions and search algorithms that improve upon previous black-box optimizations.

First of all, it is important to emphasize that the paper makes a novel and timely contribution to the code-generation for tensor programs. There has been two major directions in code-generation. One being the black-box optimization building on TVM or TVM-like templates. Another being the auto-scheduling techniques that aim to minimize human intervention in devising the templates. The abstraction introduced in the paper initiates a third thrust of probabilistic programs. It seems also important to note that it builds on TVM.

The paper enables a stochastic search space construction which the contributions are straight-forward. Especially, Figure 3 clearly demonstrates an example. However, it would assist readers better if there could be some table that enumerates the additional methods used in the probabilistic program such as Sample-Tile and so on. Also, it would be good to add an example that contrasts different methods for tensor program generation.

One issue that I have with the paper is that unlike AutoTVM for instance, it does not outline the optimization details. For example, it is not intuitive as how to incorporate pre-trained cost-models from datasets such as TenSet. It seems that lack of these information in this respect leaves readers with lots of questions. How do we incorporate pre-trained cost-models? Is the “Learning-driven search” part exactly equivalent to the AutoTVM and its variants?, In that case doesn’t that make transfer learning more difficult as the search spaces become more heterogeneous? While these are just some hypothetical questions that I enumerated, adding more details to the Section 4 would prevent the confusion.

Regarding evaluation, it seems that the largest benefits come from (1) multi-level tiling and (2) use tensor cores. However, reasons behind this nor insights were shared with the readers. This seems to make it difficult for readers to gain more insights as to how to optimize the search.

Lastly, the paper is focused on CPUs and GPUs. Considering the explosion of number of accelerators in the recent decade, it would be interesting to see how this can be incorporated into compilation for these diverse set of accelerators.

In sum, despite the above issues, the paper makes in important contribution that would enable interesting works on search and optimization. To this end, I would like to recommend the paper for a Weak Accept. However, with more concreteness to the paper addressing the issues above, I would be willing to reevaluate.

**Questions:**

* Can you fortify the Section 4?
* Would this new abstraction require changes over prior works on black-box optimization given the probabilistic programs would render much diverse set of search spaces?
* How can transfer learning in AutoTVM be enabled? Also, datasets such as TenSet?
* How would the Figure 8 look like with some fused operations such as Conv + Pool + Relu? Could you share some results if possible with some of the common patterns in DNNs?

**Ethics Review Area:**

["I don’t know"]

**Strengths And Weaknesses:**

+ timely contribution of probabilistic programming to tensor program compilation
+ abstraction proposed in Section 3 can instigate interesting research in optimization that would benefit the readers.

- unclear on details in Section 4 about learning
- unclear on how it relates to prior works in more detail (not just high level statements in 3.3)

---

> ### Author Response · Authors · 2022-08-02
> **Thank you for your helpful reviews and inputs**
>
> Thank you for your helpful reviews and inputs. We would like to clarify some of the questions below:
>
> > Table about key primitives
>
> Thank you for your suggestions. We agree that it is helpful to have a table explaining the key primitives and will include it in the final version.
>
> > Relation to prior works
>
> Thank you for your input. We agree that more examples can help here.  We also plan to open source our code, which includes SpaceCraft programs that reproduce the existing Tensor Program Optimization space in 3.3.
>
> > incorporate pre-trained cost-model
>
> Pre-trained cost models can be supplied as the cost evaluation function $\hat{f}(.)$ in Figure 7. Thank you for bringing this up! We plan to add a discussion about how cost models are applied to our flow.
>
> > transfer learning and tenset
>
> TenSet, as a dataset, could be naturally migrated to SpaceCraft because its sampling space is a small subset of what SpaceCraft could represent. We can leverage cost models transferred from existing ones like TenSet. As the search space grows, there is indeed more pressure on cost model generalization.
>
> > Sec 4
>
> Thank you for your valuable suggestion, and they are helpful in making the paper more accessible. We plan to add discussions on how to incorporate cost models and how search space puts requirements on cost model generations as listed in the above two pts.
>
> Additionally, we also plan to open source the code that allows others to plugin their own cost model and search to facilitate future research in this area.
>
> > discussions in evaluation
>
> Thank you for your suggestion, we will add discussion about how the modules affect performance, in particular the reduction in memory transfer and use of hardware accelerations.
>
> > support for a diverse set of accelerators.
>
> This is an extremely valuable point! Accelerator is indeed within the scope of our consideration. The probabilistic transformations provide a modular approach for ourselves and accelerator providers to provide specific optimization techniques. We choose to evaluate on Nvidia’s TensorCore as it contains the same set of challenges that some of the accelerator exhibits, namely the specialized tensor instructions and memory scope – it serves as a good proxy on how we can generalize to other accelerators as well.
>
> We also agree that it is interesting to see more uses of the approach to support accelerators, and would love to see follow-ups as we open source.
>
> > require changes over prior works on black-box optimization
>
> SpaceCraft allows us to generate a diverse search space controlled by the set of transformation modules. We anticipate it to compose well with black-box optimization methods. Of course, as the specified search space becomes more diverse, there can be more demands on cost models and search. This paper provides a foundational tool for us to explore tradeoffs by easily constructing different search spaces.
>
> > How would the Figure 8 look like with some fused operations such as Conv + Pool + Relu
>
> Thank you for your suggestion. Figure 8 includes some sub-graph workloads, e.g. CBR is fused conv-batchnorm-relu . We anticipate the Conv + Pool + Relu to look similar to Figure 8. The end-to-end evaluation(Figure 9) also resulted from optimizing fused subgraph workloads.

---

> > ### Comment · Reviewer_zFyW · 2022-08-09
> > **Thank you for the detailed response**
> >
> > I would like to follow up on the following:
> >
> > > > Relation to prior works
> > >
> > > Thank you for your input. We agree that more examples can help here. We also plan to open source our code, which includes SpaceCraft programs that reproduce the existing Tensor Program Optimization space in 3.3.
> >
> > > > transfer learning and tenset
> > >
> > > TenSet, as a dataset, could be naturally migrated to SpaceCraft because its sampling space is a small subset of what SpaceCraft could represent. We can leverage cost models transferred from existing ones like TenSet. As the search space grows, there is indeed more pressure on cost model generalization.
> >
> > While the people familiar with the line of research (TVM, Ansor, SpaceCraft) would understand that this may be the case, it seems that how one space would be a subset of another is not easy to follow. I do understand that the empirical results demonstrate this. However, could you elaborate on how this may be? Maybe with some example in the paper or the appendix?
> >
> > Given the promise of the authors for the requests I made in the original review, I would be happy to see the paper in the main conference.

---

> > > ### Author Response · Authors · 2022-08-09
> > > **Thank you for your helpful reviews and inputs**
> > >
> > > Thank you for your valuable suggestion, and they are very helpful in making the paper accessible to broader audiences!
> > >
> > > The relation can be further clarified as follows:
> > > AutoTVM's search space is template-based with tuning knobs of loop variables. We can declare these tuning knobs as random variables of stochastic transformations and reproduce any search space AutoTVM has.
> > >
> > > Ansor's search space is based on a fixed set of sketch generation steps, which examine the program state (such as reuse) and propose a set of possible program transformations. SpaceCraft presents a natural programming model for non-experts to build these generation steps as transformation modules (as the example in Figure 4), and we can easily grow the search space beyond the original set (as demonstrated in TC in evaluation).
> > >
> > > We plan to add these clarifications to sec 3.3. We also plan to add code examples of spacecraft to the appendix showing how the search space of AutoTVM and Ansor can be resembled by SpaceCraft in the appendix.

---

### Official Review · Reviewer_AcTZ · 2022-07-12

**Rating:** 6
**Confidence:** 4
**Soundness:** 3 good
**Presentation:** 3 good
**Contribution:** 3 good

**Summary:**

The paper outlines an approach to streamlining the process of tensor program optimization for different hardware substrates, by integrating a prior over optimization approaches. The latency of the compiled program (or a prediction thereof) is added to the prior probability of the optimization transforms as a potential function for random walk metropolis hastings, to explore alternative optimization approaches. Adding hardware specific optimizations is feasible and unlocks performance improvements on modern DL models, relative to TVM and PyTorch.

**Questions:**

How easily is the new system integrated with existing ML frameworks? Can we easily write code in PyTorch or JAX that calls into this system? Does optimization need to be re-run with every process startup, or is there a cache of previous work maintained?

How important is the prior, and how is it specified? None of the examples suggest the user can provide the prior probability of selecting any given categorical decision or relative probabilities of different transformation modules being applicable. So, is a uniform prior over categories assumed?
Do you need to scale the prior relative to the runtime "energy"?

It seems like some discussion of Bayesopt / RL based approaches would be in order. I am surprised if noone has tried RL / contextual bandits in this space, but even if that's the case it's worth calling out as future work.

It seems like the system supports only categorical/discrete decisions. Are there cases in compiler optimization where a continuous random variable is useful? If we had such a case would it be easily supported by SpaceCraft?

Do the programs produced yield interesting numerical issues? Sometimes, reordering arithmetic operations such as sum reductions can result in catastrophic cancellations.

**Limitations:**

Main limitation discussed is invalid output programs (OOM, etc), and validation being a viable approach to rectify this.
It'd be interesting to learn how long optimization takes, possibly see curves of "best runtime by # steps".

**Strengths And Weaknesses:**

Originality
The approach seems to be a novel one, and contributes to an important field of compiler optimizations for ML models. These systems facilitate performant deployment of models from training devices (GPU, high end CPU) to inference endpoints (mobile CPU, GPU). Previous work in this area could include Tensor Comprehensions, TVM, Halide, and beyond.

Quality
The "user experience" of specifying kernels in the probabilistic DSL looks a little bit awkward. An upside, as a python-embedded traced DSL, we can interleave host language control flow to study conditions such as spatial-vs-reduction. A downside is that the resulting kernel code may need to be specialized to particular tensor shapes and ranks. It also appears (but I could have misunderstood?) that transformation modules are specific to tensor programs, which could limit the applicability to performance-critical code segments. The decision to use MCMC to sample this space is an interesting one, because I would expect the outcomes to be highly nonlinear relative to the decisions being sampled. A non-smooth posterior is a nightmare for most MCMC samplers, and I'd wonder whether there's any value to MCMC relative to some kind of prior-regularized evolutionary approach (or is that basically how we are viewing the MH work here?). Do you need to scale the prior relative to the runtime "energy"?

Clarity
This is a "systems" type of paper, where observing code and before/after different optimizations is probably an effective mode of presentation, but is necessarily limited by the page limit. The "kinds of things" that can be done with SpaceCraft are reasonably well presented, but I did not feel that the "way those things are done" was very clearly communicated. Also unclear was the time spent exploring for optimizations relative to a deterministic system like TVM.

Significance
~2x speedup relative to PyTorch on BERT-Large seems significant, but without very-convenient tie-in to major ML systems may not be very usable by the community. The claim on 278-80 about ease of adding a novel TC-specific module are enticing, but probably deserve to be fleshed out further in the body of the paper rather than supplementary.

---

> ### Author Response · Authors · 2022-08-02
> **Thank you for your review and helpful inputs**
>
> Thank you for your review and helpful inputs, they would certainly help us to make the paper more accessible. We would like to clarify some of the questions below:
>
> > user experience of kernel specification
>
> We agree that it is prohibitive to specify each kernel manually. We show the tensor program together with probabilistic transformations to simplify our presentation. In practice, the generation of the original program is separated from the probabilistic transformations. We generate the original set of loops from computational graphs obtained from common machine learning frameworks such as TensorFlow and PyTorch. These generation processes are generic to shape/rank. Thank you for bringing this up. We will highlight it in our future version.
>
> > transformation modules are specific to tensor program
>
> Transformation modules are generic to tensor programs and thanks to the flexibility of the probabilistic programming model. As an example, the Multi-level-tiling module in Figure 4 can be applied to conv2d, matmul, conv1d, conv3d, and other related patterns.
>
> The generic module analyzes the common patterns among these tensor programs, such as whether or not a computation contains a reduction that can bring memory reuse, and makes probabilistic recommendations about possible choices. As a result, the transformation module combining analysis and stochastic decisions can be super powerful and serve as a generic way to generate rich search space informed by domain knowledge.
>
> Thank you for bringing this point up, and we will revise our paper to highlight this fact.
>
> >  Choice of MCMC and evolutionary search
>
> Thank you for your suggestions. We formalize the problem as MAP to inform the generic framework design. We did also support and use evolutionary search, which can be viewed as a way of parallel chain MCMC for MAP inference. We did find that some stochasticity in the overall process helped explore the space. We will make it clear in the updated version.
>
> > time spent exploring for optimizations relative to TVM
>
> TVM also employs automatic scheduling and search. We made the orthogonal contributions on search space construction that can enhance these previous approaches as well.
>
> > more details about novel TC-specific module in the paper
>
> Thank you for your suggestion. We left it out due to the space limit. We believe it is good to stay at general approach level and will summarize and discuss key elements in the TensorCore module to give the readers more insights into it.
>
> > Integration with PyTorch or JAX
>
> Our approach interpolates with ML frameworks. We can ingest PyTorch models and use them to construct the original loop programs, which then get optimized by transformation modules and search. We did cache cost models and previous runs so there is no need to re-run the process. We also plan to add JAX support. We plan to open source the code that reproduces these steps.
>
> > Prior specification
>
> We find that it is important to specify the “structural priors”, e.g. when there are memory reuse opportunities, try out tiling the loops and possibly parallelization. But the sizes can be left as uniform.
>
> > Bayesopt / RL
>
> Thank you for your suggestion. The search space construction is orthogonal to search approaches taken, so we expect our approach can also empower future research on BayesOpt/RL for tensor program optimization by modular search space construction. We will add a discussion.
>
> > continuous random variable
>
> Our system can easily support continuous random variables. So far we have not yet found a direct use case of continuous random variable, but we would love to see possible use cases of it in the future.
>
> > numerical issue
>
> We have not experienced numerical issues, primarily likely due to the fact that model neural network workloads operate in a good input space. Note that similar kinds of optimizations are performed by libraries like cuDNN.

---

### Author Response · Authors · 2022-08-08
**Paper is revised to reflect reviews and discussion**

We would love to thank all the reviewers again for sharing your insights. The paper draft has been updated to reflect our discussion.

Summary of updates:
- Table of all key transformation primitives in SpaceCraft (Appendix A.7);
- More details on experiment setting, state-of-the-art, and baseline choices (Section 6.1);
- TensorCore and effort to add new hardware backends (Section 6.3 and Appendix A.4)
- Discussion about cost model, pre-training, and TenSet (Section 4);
- Mentioning Bayesian optimization and reinforcement learning can be used as the search algorithm (Section 4);
- User experience and integration with ML frameworks (PyTorch/JAX/TensorFlow) (Section 3.1 and Appendix A.6);
- Tuning time is comparable with baselines (Section 6.2, Appendix A.5);
- Clarification that transformation modules are generic to tensor programs (Section 3.2);
- Clarification of the dependency between program structure and previous sampling decisions (Section 3.1);
- Clarification that Figure 8 includes subgraph-level comparison (Section 6.1);

---

### Meta-Review · Area_Chair_j1L7 · 2022-08-26

**Recommendation:** Accept
**Confidence:** Certain

**Metareview:**

With many of the changes proposed by the authors in response to reviewer feedback, I think this will be a very good paper, arguably warranting even higher scores than the updated reviewer scores would indicate, which are already unanimously in favor of acceptance post-rebuttal. The performance improvements obtained here seem very significant, and I think that this paper represents an enormous engineering effort, which I think has potentially enormous value to researchers and practitioners training large end-to-end deep learning models. Despite the paper's significant focus on engineering and systems concerns, I think the potential value to machine learning here is quite clear, and therefore clearly a fit for NeurIPS.

**Award:**

No

---

### Decision · Program_Chairs · 2022-09-14

Accept